# Modulation of antigen discrimination by duration of immune contacts in a kinetic proofreading model of T cell activation with extreme statistics

**Jonathan Morgan[1,2], Alan E. Lindsay[1]***

**1** Department of Applied and Computational Mathematics and Statistics, University of Notre Dame, South Bend, Indiana, United States of America, **2** Biophysics Graduate Program, University of Notre Dame, South Bend, Indiana, United States of America

* a.lindsay@nd.edu

**Data Availability Statement:** This work does not involve data. Our code is available at https://github.com/jmorga15.

## Abstract

T cells form transient cell-to-cell contacts with antigen presenting cells (APCs) to facilitate surface interrogation by membrane bound T cell receptors (TCRs). Upon recognition of molecular signatures (antigen) of pathogen, T cells may initiate an adaptive immune response. The duration of the T cell/APC contact is observed to vary widely, yet it is unclear what constructive role, if any, such variations might play in immune signaling. Modeling efforts describing antigen discrimination often focus on steady-state approximations and do not account for the transient nature of cellular contacts. Within the framework of a kinetic proofreading (KP) mechanism, we develop a stochastic *First Receptor Activation Model* (FRAM) describing the likelihood that a productive immune signal is produced before the expiry of the contact. Through the use of extreme statistics, we characterize the probability that the first TCR triggering is induced by a rare agonist antigen and not by that of an abundant self-antigen. We show that defining positive immune outcomes as resilience to extreme statistics and sensitivity to rare events mitigates classic tradeoffs associated with KP. By choosing a sufficient number of KP steps, our model is able to yield single agonist sensitivity whilst remaining non-reactive to large populations of self antigen, even when self and agonist antigen are similar in dissociation rate to the TCR but differ largely in expression. Additionally, our model achieves high levels of accuracy even when agonist positive APCs encounters are rare. Finally, we discuss potential biological costs associated with high classification accuracy, particularly in challenging T cell environments.

## Author summary

Physical contact between the T cell and antigen presenting cell (APC) is essential for productive immune signaling. Wide variations in this contact time have been observed yet little is known of mechanisms controlling this crucial timescale, nor how its duration may impact antigen discrimination. We develop and analyze a probabilistic mathematical

**Funding:** A.E.L acknowledges funding under NSF DMS 1815216. The funders had no role in study design, data collection and analysis, decision to publish, or preparation of the manuscript.

**Competing interests:** The authors have declared that no competing interests exist.

model of T cell activation which combines kinetic proofreading (KP) with a finite contact duration. Our model is capable of suppressing large populations of self ligands while remaining sensitive to only a single agonist in T cell/APC cellular contacts. Additionally, we explored two challenging cases, one in which self and agonist antigen are similar and one in which agonist positive APCs are rare. We found that our model could overcome these environmental challenges by increasing the number of kinetic proofreading steps. Finally, we discuss the potential biological costs of achieving such accuracy. Our work demonstrates the extreme effectiveness of kinetic proofreading in a temporal context while also demonstrating the possible challenges in biological implementation of such a model.

## Introduction

### Background

T cells are immune cells that continuously search for molecular signatures (antigens) of pathogens and upon recognition, can initiate an adaptive immune response. When a T cell encounters an antigen presenting cell (APC), T cells recognize antigen through binding of their T cell receptors (TCR) to the peptide-MHC complex (pMHC) on the APC. For an efficient immune response, T cells must be able to recognize when an APC is agonist positive, where an APC displays foreign antigen indicating an immune response is appropriate, or agonist negative, where an APC displays only self antigen indicating no response is the appropriate action. This recognition is accomplished at the level of receptor/ligand interactions, where TCR/pMHC binding can result in the generation of an intracellular signal, such as $Ca^{2+}$ influx, that may lead to T cell activation [1]. This process of antigen recognition is marked by rapid recognition speeds with single-molecule sensitivity to agonist ligands [2]. For example, CD4[+] T cells can exhibit $Ca^{2+}$ signaling when stimulated with a single molecule of foreign antigen [3, 4]. Similarly, CD8[+] T cells have been shown to recognize as few as 3 molecules of a foreign antigen [5]. This single-digit molecule sensitivity is particularly remarkable given how short-lived TCR/pMHC interactions can be [6–8]. In addition, T cells can amplify small differences in antigen affinity into large differences in their responses [9–15]. The combination of these features of T cells have led many to characterize T cell antigen discrimination as being near-perfect [16–22], where the T cell is capable of recognizing agonist positive APCs (APCs with at least one agonist antigen on the surface) and simultaneously remain non-reactive to agonist negative APCs (APCs with no agonist presence) even when agonist populations are small, self antigen populations are large, and the distinguishing characteristic between agonist and self antigen is a slight difference in TCR affinity. Moreover, some studies have shown that TCRs may be more specifically tuned to the antigen dissociation rate, rather than the affinity [6, 11, 23–27]. This may indicate that T cells are specifically tuned to recognize agonist and self antigen primarily based on the antigen dissociation rate with the TCR.

### Kinetic proofreading

The observed features of T cell antigen discrimination led McKeithan to introduce a kinetic proofreading (KP) model (Fig 1) for TCR activation. KP proposes that a productive immunological signal can arise after the completion of several intermediate conformational changes, all while being subject to TCR/pMHC disassociation that may eliminate progress towards activation [28]. KP has since become a prominent and highly studied conceptual framework for

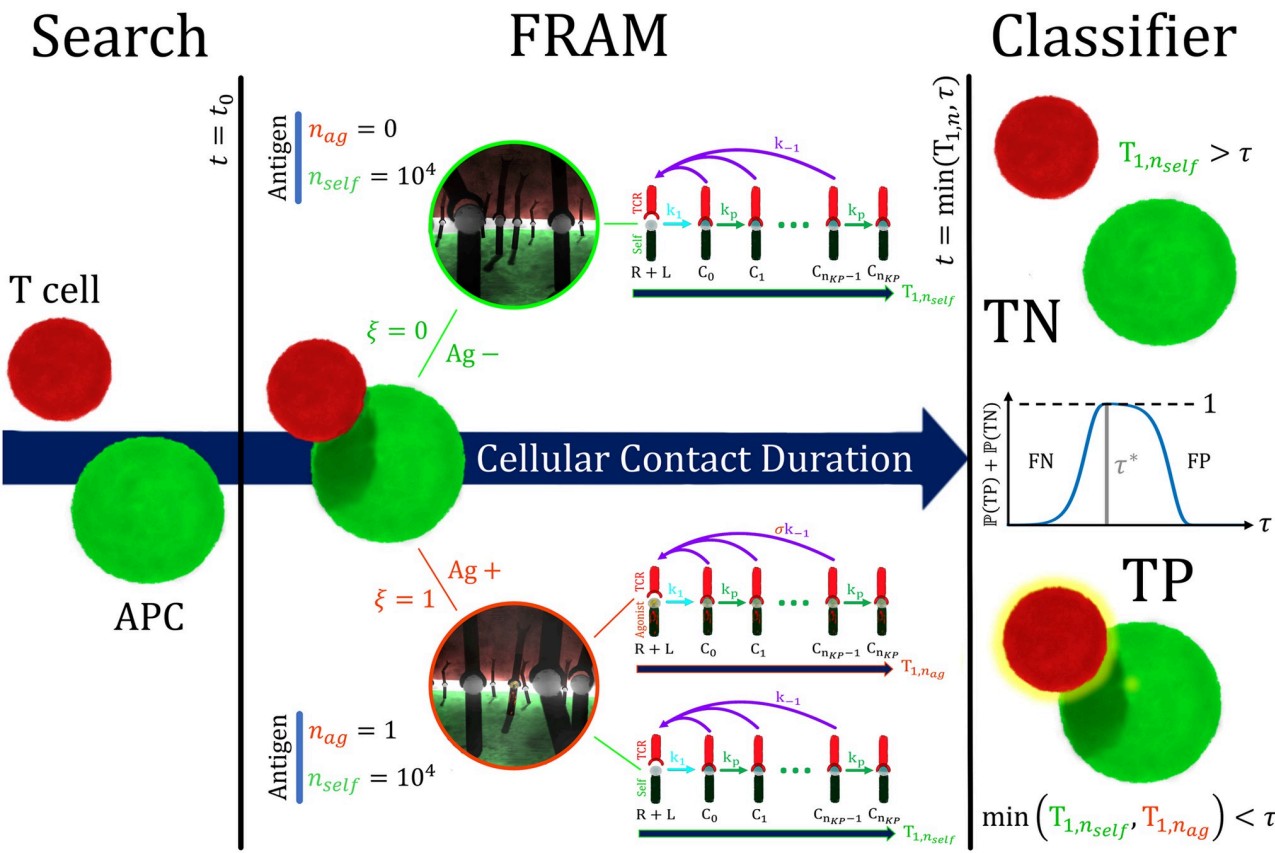

**Fig 1. The First Receptor Activation Model (FRAM) as a classifier of APC status $\xi \in \{0, 1\}$.** The T cell and APC form a cellular contact at $t = t_0$. An APC is agonist positive ($\xi = 1$) with probability $\mathbb{P}(\xi = 1) = \rho_{ag}$ and agonist negative ($\xi = 0$) with probability $\mathbb{P}(\xi = 0) = 1 - \rho_{ag}$. If $\xi = 1$ then there is $n_{ag} = 1$ agonist antigen and $n_{self} = 10^4$ self antigen. Activation is the event ($\min(T_{1,n_{self}}, T_{1,n_{ag}}) < \tau$) which results in a true positive (TP) classification. If $\xi = 0$ then correct classification occurs when none of the self antigen activate ($T_{1,n_{self}} > \tau$) which results in a true negative (TN). We measure the accuracy ($\mathbb{P}(TP) + \mathbb{P}(TN)$) of the FRAM as an APC classifier with respect to the cellular contact duration ($\tau$). Cellular contacts of too short duration result in a high false negative rate while overly long contacts return a high false positive probability, with both scenarios reducing accuracy. Decision accuracy is maximized at fixed contact duration ($\tau^*$). Published with permission under the CC BY 4.0 licence.

understanding antigen discrimination [29–34]. Steady state analysis of KP models shows that the mechanism is certainly capable of ligand discrimination, however, large increases in the number of intermediate states would increase the specificity of the model, but only at the expense of model sensitivity to agonists [4, 35], i.e., the KP mechanism could reproduce observed ligand discrimination only when the agonist population was sufficiently large. This is because the increase in intermediate bound states would yield an unacceptably low steady state fraction of activated TCRs when there were only few agonist antigen. This result calls into question the feasibility of KP models being the sole descriptor of the observed antigen discrimination in T cells [16–22, 36], since both, high specificity and sensitivity to small populations of agonists, have been observed in experiments. Nevertheless, despite the shortcomings of theoretical models, KP is supported by observations that show the antigen dissociation rate is a significant indicator of self or agonist antigen, and experimental work has continued to show results that indicate kinetic proofreading is involved in the antigen recognition process [34, 37, 38].

One physical realization of the KP mechanism in TCR activation would be the phosphorylation of the immunoreceptor tyrosine-based activation motifs (ITAMs) of the TCR-CD3

complex. The authors in [39] showed that all six ITAMs on the CD3$\zeta$ chain are phosphorylated in a specific, sequential manner and is a potential requirement for TCR activation. This could be akin to a 6-step kinetic proofreading mechanism for TCR activation ($n_{KP}$ = 6). However, more recent evidence indicates that the number of KP steps may be significantly smaller. In the work by Voisinne et al. [40], it was observed that phosphorylations of the CD3 chains of the TCR-CD3 complex, and the recruitment of ZAP-70, were similar regardless of the differences in antigen affinities. Furthermore, the authors found that the most significant regulatory step, associated with antigen affinity, was the phosphorylation of two sites on ZAP-70 itself. This may reveal that the majority of the phosphorylation events in the TCR activation process occur in a similar manner regardless of the type of antigen presented to the TCR and that a smaller $n_{KP}$ may be more appropriate. Indeed, Pettmann et al. [41] used the concept of a discrimination power ($\gamma$) to approximate the effective number of KP steps in T cell activation. The authors measured $\gamma \approx 2.7$, and argued that just 2 or 3 KP steps could effectively explain the discrimination power seen in their experiments.

## Cellular contact times

It has been estimated that APCs encounter 500–5000 T cells every hour (dendritic cells) [42–44], suggesting many T cell/APC contacts are formed over short periods in time. Experimental work has shown evidence that the duration of these contacts may be significant in antigen recognition [45–51] and some studies have classified the engagement period into distinct phases based on cellular contact times [52–57]. The first (phase I) of these periods may involve multiple, short-duration, transient encounters between T cells and APCs which persist until a certain threshold of accumulated signal followed by a second period (phase II) in which a long, stable contact forms. In the context of a KP mechanism of activation, it is clear that a sufficiently long-lasting cellular contact is required in order for a productive immune signal to be generated and so early termination of the cellular contact may hypothetically prevent T cell activation [58]. In addition, early termination of a T cell/APC contact could prevent the transition from a short transient contact to a more stable contact. Together, this suggests that the duration of the T cell/APC contact may have an important role in antigen discrimination.

When the duration of the T cell/APC contact is explicitly modeled, the previously employed steady-state analysis may not be appropriate [28, 30, 36, 59, 60]. For example, if the cellular contact duration is shorter than the timescale of relaxation to the steady state, the equilibrium configuration may not be attained. Similarly, in scenarios where activation is defined by an accumulation of productive signal, integration over periods of non-equilibrium behavior results in a non-trivial dependence of contact duration [61]. Finally, in rebinding models where multiple short TCR/antigen engagements [7, 17, 19, 35] can lead to rare TCR triggering events, the role of a finite cellular contact duration may be a significant factor in modeling immunological outcomes.

## Summary of results

In this paper, we develop a *First Receptor Activation Model* (FRAM) which describes T cell activation as the event when any one receptor is triggered by an antigen ligand before the cellular contact expires. For a population of APCs where a fraction ($\rho_{ag}$) are agonist positive, we determine the accuracy of a T cell as an APC classifier, which is the probability $\Gamma$ that a T cell will correctly classify a randomly chosen member of this population. Mathematically, this involves calculating certain extreme statistics [62–65] due to the fact that while a single T cell/APC interaction generates numerous independent TCR/pMHC reaction pairs, it is the fastest of these that will set the activation time. A positive outcome requires our model to be reactive

to a small number of agonist antigen while non-reactive during numerous interactions with abundant self-antigen.

We observe that the accuracy of our model varies considerably with contact duration $\tau$ and demonstrate that there often exists a window of time such that $\Gamma \approx 1$, i.e. the accuracy is near-perfect. We find that regions of high accuracy expand/contract as the number $n_{KP}$ of KP states increases/decreases. A similar relationship is observed with respect to variations in the ratio $\sigma$ between dissociation rates of self and agonist antigen (Fig 1). Specifically, classification accuracy is higher at larger $\sigma$ values and decreases as $\sigma \to 1^{+}$. We show that our model can achieve near-perfect classification accuracy for just a single agonist antigen $n_{ag} = 1$, $n_{self} = 10^{4}$ self antigen, and a relative dissociation factor $\sigma = 2$. This demonstrates that by increasing the number of kinetic proofreading steps the FRAM is able to overcome the challenge of remaining sensitive to a single angonist antigen, and suppressing receptor triggering for a large self population of antigen, even when self and agonist antigen appear similar in the lens of a KP mechanism.

The proportion $\rho_{ag}$ of agonist positive APCs influences the difficulty of the classification task. This is especially so when $\rho_{ag}$ is small since this means most cellular contacts are made of agonist negative cells, and the suppression of large self antigen populations becomes more significant. When agonist positive cells are rare, we found that shorter-duration contacts decrease the likelihood that a T cell activation is a result of a false positive. This demonstrates that the agonist positive prevalence may influence optimal cellular contact durations. However, our results also demonstrate that a sufficient number of KP steps could overcome this challenge by maximizing the accuracy of the T cell for all $\rho_{ag}$ and effectively making the optimal cellular contact durations independent of the agonist positive prevalence.

Lastly, we quantify a potential biological constraint in the FRAM. We assume that forward KP reactions involve events with an energy cost [1, 28, 66–68], which we call *futile* reactions if the TCR/antigen dissociate before TCR activation. As mentioned, our results show that increasing $n_{KP}$ could overcome the challenges of similar self/agonist ligands, large differences in self/agonist expression, and rare agonist positive APCs. However, we show that the increased number of KP steps results in either longer-lasting cellular contacts or faster KP rates, both of which cause an increased number of futile reactions. We conclude that accurate antigen discrimination may come with associated energetic costs.

## Model

We consider a population of APCs, each of which is agonist positive ($\xi = 1$) with probability $\mathbb{P}(\xi = 1) = \rho_{ag}$ or agonist negative ($\xi = 0$) with probability $\mathbb{P}(\xi = 0) = 1 - \rho_{ag}$. In the case $\xi = 1$ there are $n_{self} = 10^{4}$ self antigen and $n_{ag} = 1$ agonist antigen in any T cell/APC contact while for $\xi = 0$ there are $n_{self} = 10^{4}$ self antigen and $n_{ag} = 0$ agonist antigen in any contact. We assume that during T cell interrogation of an APC surface, the TCR density is sufficient to engage all antigen in the contact. Each TCR/antigen pair undergoes dynamics according to the KP mechanism Fig 1 and the first receptor activation model (FRAM) defines T cell activation as the event when any TCR reaches the signaling state $C_{n_{KP}}$.

We define $T_{1,n_{ag}}$ to be the signaling time of an agonist antigen and agonist activation to be the event ($T_{1,n_{ag}} < \tau$). Self activation of the T cell occurs when one of the $n_{self} = 10^{4}$ self antigen [14] activate a TCR before the cellular contact expires ($T_{1,n_{self}} < \tau$). We emphasize that the self activation time is $T_{1,n_{self}} = \min\{t_{1}, \ldots, t_{n_{self}}\}$ where $t_{i}$ is the activation time of the $i^{th}$ signaling pair and is hence an example of an extreme statistic [62–65, 69]. We remark that in contrast to standard problems in extreme statistics, the times $\{t_{j}\}_{j=1}^{n_{self}}$ are not i.i.d. due to receptor and ligand binding kinetics. However, we show that under some conditions the results on the

extreme statistics involving a large number of i.i.d. samples still apply (Sec. 2 of S1 Text). It is also worth noting that extreme events are typically very fast relative to individual first passage events, but productive T cell activation requires that $T_{1,n_{ag}} \ll T_{1,n_{self}}$.

We derive an analytical expression for the first passage time $T_{1,n_{ag}}$ of TCR activation in the case of a single TCR/pMHC pair (Sec. 1.1 of S1 Text). For a large population of ligands and receptors, we approximate the distribution for the extreme statistic $T_{1,n_{self}}$ by introducing a system of ordinary differential equations (Sec. 1.2 of S1 Text). In addition, we connect the limiting case $n_{self} \to \infty$ with recent results of extreme statistic for continuous time Markov chains with discrete states [69]. Specifically, as $n_{self} \to \infty$, we confirm (Sec. 2 of S1 Text) the limiting behavior

$$T_{1,n_{self}} \approx_d \text{Weibull}((An_{self})^{-\frac{1}{n_{KP}}}, n_{KP}), \qquad A = \frac{k_p^{n_{KP}}}{n_{KP}!}, \tag{1}$$

where a random variable $X \geq 0$ has a *Weibull* distribution with scale parameter $\lambda > 0$ and shape parameter $k > 0$ if $\mathbb{P}(X > x) = e^{-(x/\lambda)^k}$. In such a case, we define $X =_d \text{Weibull}(\lambda, k)$.

In the event that $\xi = 1$, both activation cases result in a true positive since the T cell activated and was in contact with an agonist positive APC. The T cell does not activate when none of the $n_{self} = 10^4$ self antigen activate a TCR before the cellular contact expires ($T_{1,n_{self}} > \tau$). When $\xi = 0$, this results in a true negative.

We utilize the FRAM as an APC classifier, where the task is to correctly identify agonist positive and agonist negative APCs. In our model, a T cell makes contact with an APC at time $t = 0$ and remains in contact till the expiry time $t = \tau$. A true positive classification occurs when an agonist positive APC is activated before contact expiry ($\xi = 1$ and $\min\{T_{1,n_{ag}}, T_{1,n_{self}}\} < \tau$)). A true negative classification occurs when a T cell is in contact with an agonist negative APC and the T cell does not activate before the expiration of the cellular contact ($\xi = 0$ and $T_{1,n_{self}} > \tau$). Mathematically, the four outcomes of the FRAM have probabilities

$$\mathbb{P}(\text{TP}) = \mathbb{P}(\min(T_{1,n_{ag}}, T_{1,n_{self}}) < \tau \mid \xi = 1)\mathbb{P}(\xi = 1), \tag{2a}$$

$$\mathbb{P}(\text{FN}) = 1 - \mathbb{P}(\text{TP}), \tag{2b}$$

$$\mathbb{P}(\text{FP}) = \mathbb{P}(\{T_{1,n_{self}} \leq \tau\} \mid \xi = 0)\mathbb{P}(\xi = 0), \tag{2c}$$

$$\mathbb{P}(\text{TN}) = 1 - \mathbb{P}(\text{FP}). \tag{2d}$$

The main parameters of interest are $\{\tau, \rho_{ag}, n_{KP}, \sigma\}$, where $\tau > 0$ is the expiry time, $\rho_{ag} \in (0, 1)$ is the agonist positive prevalence, $n_{KP}$ is the number of KP steps, and $\sigma > 1$ is the ratio of TCR/antigen dissociation rates between self and agonist antigen which is referenced in previous experimental and theoretical works [28, 30, 61, 67, 70]. Finally, we make the assumption that the first receptor activation by self antigen is independent of the first receptor activation by agonist antigen, i.e.,

$$\mathbb{P}(\min(T_{1,n_{ag}}, T_{1,n_{self}}) < \tau) = \mathbb{P}(T_{1,n_{ag}} \leq \tau) + \mathbb{P}(T_{1,n_{self}} \leq \tau) - \mathbb{P}(T_{1,n_{ag}} \leq \tau)\mathbb{P}(T_{1,n_{self}} \leq \tau).$$

Since we are modeling the self and agonist situations separately in the agonist positive T cell/APC contacts, then independence is guaranteed. However, this is clearly an approximation

**Table 1. A list of model parameters and variables with their default values or ranges (unless otherwise specified).**

| Symbol | (Unit) | Range | Description |
|---|---|---|---|
| \multicolumn Variable ranges and descriptions. | | | |
| $k_1$ | $(s^{-1})$ | 1.0 | TCR/pMHC binding rate |
| $k_P$ | $(s^{-1})$ | 1.0 | TCR complex forward (phosphorylation) rate |
| $k_{-1}$ | $(s^{-1})$ | 1.0 | TCR/pMHC dissociation rate |
| $n_{KP}$ | None | $\{3, 10\}$ | Number of KP steps |
| $\rho_{ag}$ | None | $(0, 1)$ | Agonist positive prevalence |
| $\sigma$ | None | $(1, \infty)$ | Self antigen dissociation rate multiplier |
| $\tau$ | $(s)$ | $(0, \infty)$ | Time of T cell/APC disengagement |
| $n_{self}$ | Antigen | $10^4$ | Population of self antigen |
| $n_{ag}$ | Antigen | $\{0, 1\}$ | Population of agonist antigen |
| $T_{1,n_{ag}}$ | $(s)$ | $(0, \infty)$ | Agonist induced TCR activation time |
| $T_{1,n_{self}}$ | $(s)$ | $(0, \infty)$ | Self induced TCR activation time |
| $\xi$ | None | $\{0, 1\}$ | Random variable denoting APC condition |
| TP | None | None | Event where $\xi = 1$ and $\min(T_{1,n_{ag}}, T_{1,n_{self}}) < \tau$ |
| TN | None | None | Event where $\xi = 0$ and $T_{1,n_{self}} > \tau$ |
| $\Gamma$ | None | $(0, 1)$ | T cell accuracy |
| $n_e$ | None | $(0, \infty)$ | Count of forward KP reactions |

since we do not account for possible interactions, or interference, between self and agonist ligands.

We explore the properties of the FRAM by quantifying the T cell accuracy

$$\Gamma = \mathbb{P}(\text{TP}) + \mathbb{P}(\text{TN}). \tag{3}$$

The accuracy shares similarities with some earlier measures of sensitivity and specificity in kinetic proofreading models [30], however, there are two main differences. First, the measure $\Gamma$ combines both sensitivity and specificity (each can be individually described through $\mathbb{P}(\text{TP})$ and $\mathbb{P}(\text{TN})$ into a single measurement. The second is that we are applying the measure at the population level of T cells, or more accurately, a population of T cell/APC contacts, and not at the level of individual receptors/ligand interactions Fig 1.

A list of parameters and variables together with meanings and ranges are given in Table 1. The derivation of the probability density describing $T_{1,n_{ag}} = T_{1,1}$ is shown in Sec 1.1 of S1 Text. In addition, we also describe in detail how the extreme statistic $T_{1,n_{self}}$ is sampled in Sec 1.2 of S1 Text. The focus of the remainder of the article is analyzing variations in the accuracy $\Gamma$ with respect to the parameters $\{\tau, \rho_{ag}, n_{KP}, \sigma\}$.

## Results

### Influence of contact duration on T cell accuracy

We explored the influence of the T cell/APC contact duration on T cell accuracy by computing $\Gamma(\tau)$ over a range of $\tau$ for several values of $\sigma$ and $n_{KP}$ (Fig 2A–2C). When $\rho_{ag} = 0.5$, we have the lower bound $\Gamma \gtrless 0.5$ indicating poor APC classification accuracy (i.e. $\mathbb{P}(\text{TP}) \approx \mathbb{P}(\text{FP})$). If $n_{KP}$ and $\sigma$ are insufficiently large, the FRAM cannot accurately classify APC contacts for any $\tau$ ($\sigma = 10^2$ in Fig 2A). However, in the cases where $\sigma$ is sufficiently large ($\sigma \geq 10^4$ in Fig 2A) and/or the number of proofreading steps is sufficiently large ($n_{KP} = 10$ in Fig 2B), a range of contact times exists where $\Gamma(\tau) \approx 1$ ($\tau \in [10^4, 10^5]$ and $\sigma = 15$ in Fig 2B).

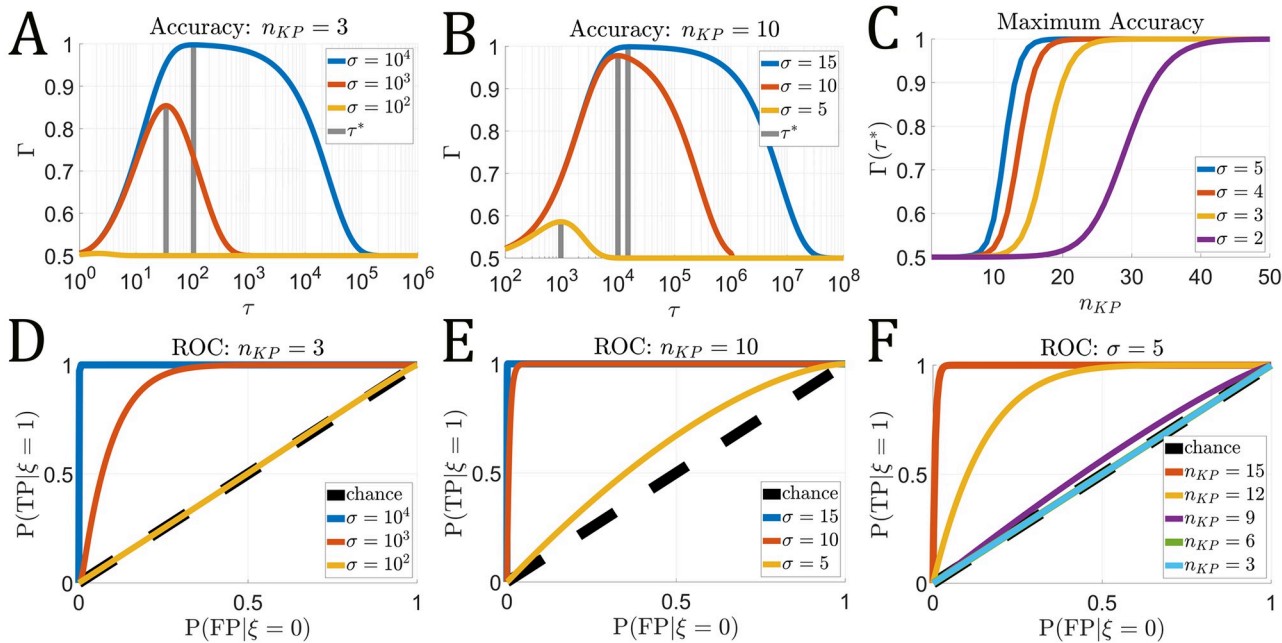

**Fig 2. T cell accuracy as contact duration $\tau$ varies.** Kinetic proofreading parameters given in Table 1. **A–B** The T cell classification accuracy $\Gamma(\tau)$ for $n_{KP} = 3$ (**A**) and $n_{KP} = 10$ (**B**). For each value of $\sigma$, the time $\tau^*$ of maximum accuracy is highlighted (gray vertical lines). **C** The maximum accuracy $\Gamma(\tau^*)$ as $n_{KP}$ varies for values $\sigma = \{2, 3, 4, 5\}$. **D–F** Receiver operating characteristic (ROC) for $n_{KP} = 3$ (**D**) and $n_{KP} = 10$ (**E**) at various $\sigma$ values. **F** The ROC for $\sigma = 5$ and varied $n_{KP}$.

To identify scenarios where effective APC classification is achieved, we determine the maximum accuracy over a range of contact durations,

$$\Gamma(\tau^*) = \max_{\tau > 0} \Gamma(\tau). \tag{4}$$

By increasing $n_{KP}$, we find that one can attain a near-perfect classification accuracy at the optimal T cell/APC contact duration ($\Gamma(\tau^*) \approx 1$), even in cases where agonist and self antigen only differ slightly in dissociation rate but differ largely in expression. Since $\Gamma(\tau^*) = 1$ if and only if $\mathbb{P}(\text{TP}|\xi = 1; \tau^*) = 1$ and $\mathbb{P}(\text{TN}|\xi = 0; \tau^*) = 1$, this indicates that the T cell always activates when in contact with an agonist positive APC and never activates when in contact with an agonist negative APC (Fig 2C).

The receiver operating characteristic (ROC) is plotted in Fig 2D–2F and allows for a more nuanced consideration of classification outcomes over the parameter $\tau$. An indicator of a classifier's performance is the area under the curve (AUC). We see that the maximum AUC (AUC $\approx 1$) can be achieved by increasing $\sigma$ (Fig 2D and 2E) and/or increasing $n_{KP}$ (Fig 2F), which indicates the model is capable of identifying agonist positive and agonist negative contacts with near-perfect accuracy. The worst-case AUC for the FRAM is when $\mathbb{P}(\text{TP}|\xi = 1) \approx \mathbb{P}(\text{FP}|\xi = 0)$ for all $\tau$. This equality also yields $\mathbb{P}(\xi = 0|\text{T}_{1,n} < \tau) = \rho_{ag}$ and $\mathbb{P}(\xi = 1|\text{T}_{1,n} < \tau) = 1 - \rho_{ag}$ for all $\tau$. This means that in these cases, the FRAM as a classifier can do no better than classifying whichever population is in the majority. We call this observation the baseline and discuss this more in-depth in the next section. Similar to our result in Fig 2C, we show that increasing the number of kinetic proofreading steps allows for arbitrary increases in the AUC, even when $\sigma$ is small (Fig 2F).

### The effects of an agonist positive prevalence

Previously we assumed an equal proportion of agonist positive and negative APCs ($\rho_{ag} = 0.5$). Variations in this prevalence can lead to significant changes in the accuracy and optimal cellular contact duration, as we now demonstrate by varying $\rho_{ag}$ between 0 and 1. In Fig 3A–3C we plot the optimal accuracy $\Gamma(\tau^*)$ against $\rho_{ag}$ for several values of $n_{KP}$ and $\sigma$. In Fig 3A, we observe at the value of $\sigma = 10^4$ that $\Gamma \approx 1$ for all $\rho_{ag}$, indicating accurate classification. As $\sigma$ decreases and/or $n_{KP}$ is too small, we find that the FRAM is incapable of accurately identifying the agonist positive or agonist negative cells for small or large $\rho_{ag}$, respectively. In these cases, the *baseline* limiting case for the accuracy becomes

$$\Gamma^{(BL)} = \left| \frac{1}{2} - \rho_{ag} \right| + \frac{1}{2}. \tag{5}$$

Eq (5) reflects the probability of an accurate T cell response with the strategy of always activating when $\rho_{ag} \geq \frac{1}{2}$ and never activating when $\rho_{ag} < \frac{1}{2}$. In Fig 3B we show results for $n_{KP} = 10$

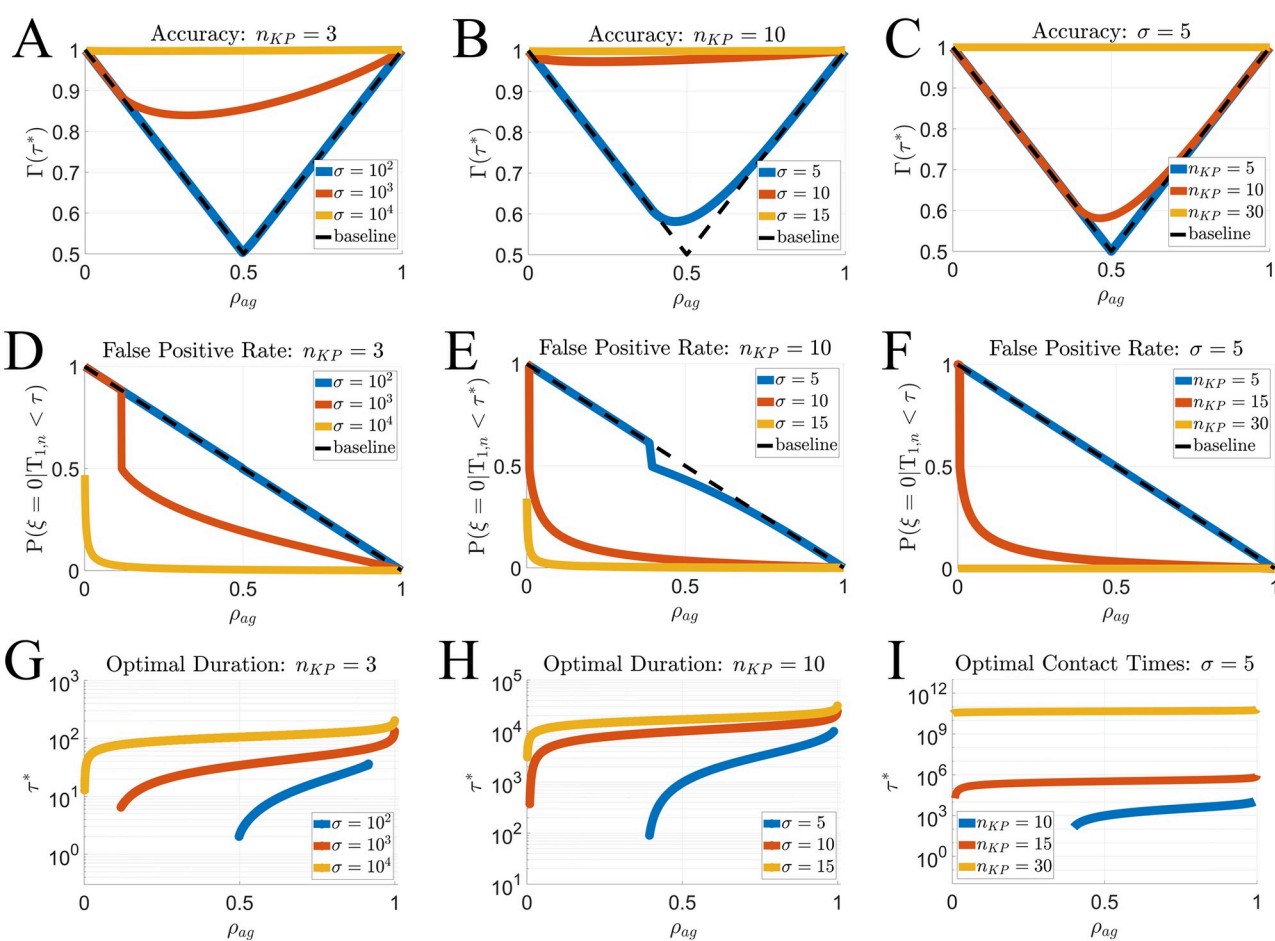

**Fig 3. Effect of agonist positive prevalence ($\rho_{ag}$) on T cell accuracy ($\Gamma(\tau^*)$) and false positive rate at the optimal contact duration (6). A–C** Accuracy $\Gamma(\tau^*)$ versus $\rho_{ag}$ for $n_{KP} = 3$ (**A**), $n_{KP} = 10$ **B**, and varied $n_{KP}$ with $\sigma = 5$ (**C**). The baseline (black dashed line) shows the worst accuracy a model can accomplish given that the optimal cellular contact duration is chosen. **D–F** False positive rate (6) at the maximizing cellular contact time ($\tau^*$). **G–I** Effect of agonist positive prevalence on the optimal cellular contact duration. Curve endpoints indicate where the FRAM reduces to baseline, i.e., $\tau^* \rightarrow 0$ if $\rho_{ag} < 0.5$ or $\tau^* \rightarrow \infty$ if $\rho_{ag} > 0.5$.

and observe higher accuracy at smaller values of $\sigma$ and over larger ranges of $\rho_{ag}$ when compared to $n_{KP}$ = 3. In Fig 3C we observe that even for small $\sigma$, perfect accuracy can be obtained for all $\rho_{ag}$ by increasing $n_{KP}$.

When agonist positive contacts are rare ($0 < \rho_{ag} \ll 1$), accurate classification becomes more challenging and the risk of false positives is greater. We use the outcome probabilities and determine the false positive rate $\mathbb{P}(\xi = 0|\mathrm{T}_{1,n} < \tau^*)$ which is defined as

$$\mathbb{P}(\xi = 0|\mathrm{T}_{1,n} < \tau^*) = \frac{\mathbb{P}(\mathrm{FP}|\xi = 0; \tau^*)(1 - \rho_{ag})}{\mathbb{P}(\mathrm{FP}|\xi = 0; \tau^*)(1 - \rho_{ag}) + \mathbb{P}(\mathrm{TP}|\xi = 1; \tau^*)\rho_{ag}}. \tag{6}$$

In Eq (6), evaluation is at the optimal cellular contact duration $\tau^*$ and $\mathrm{T}_{1,n}$ is the activation time of the FRAM with an unknown APC condition. As $\rho_{ag}$ decreases, the probability that any T cell activation is a false positive increases (Fig 3D–3F). Again, as accurate APC identification becomes too difficult, we identify convergence to the baseline case

$$\mathbb{P}(\xi = 0|\mathrm{T}_{1,n} < \tau^*) = 1 - \rho_{ag}. \tag{7}$$

In Fig 3D and 3E, we demonstrate that increasing $\sigma$ can greatly reduce the likelihood of false positives. However, even the cases that were sufficient to yield nearly 100% accuracy ($\sigma = 15$ in Fig 2B), still yield a high false positive rate when $\rho_{ag}$ is small ($\sigma = 15$ in Fig 3E). However, in Fig 3F) we choose $\sigma = 5$ and show that increasing $n_{KP}$ can completely inhibit the false positive rate ($\mathbb{P}(\xi = 0|\mathrm{T}_{1,n} < \tau^*) \approx 0$) for nearly all $\rho_{ag}$.

In the FRAM, a decrease in cellular contact duration serves to decrease the probability of activation in any T cell/APC encounter. The curves in Fig 3G–3I represent ranges of $\rho_{ag}$ where the FRAM has some ability to classify both APC conditions, not just the majority. The endpoints of these curves represent the transition to the baseline accuracy (4) where $\tau^* \to \infty$ if $\rho_{ag} > \frac{1}{2}$, and $\tau^* \to 0$ if $\rho_{ag} < \frac{1}{2}$. As $\rho_{ag}$ decreases, the false positive rate increases. In light of this, we observe that $\tau^*$ decreases with $\rho_{ag}$ (Fig 3G–3I). Increasing $\sigma$ (Fig 3G and 3H) reduces the influence of $\rho_{ag}$ on the cellular contact duration and decreases the range of $\rho_{ag}$ where accuracy reduces to the baseline (4). In Fig 3I, we note that the contact duration becomes nearly independent of $\rho_{ag}$ as $n_{KP}$ is increased.

Taken together, these results suggest that the accuracy is independent of the agonist positive prevalence when $\sigma$ and/or $n_{KP}$ is sufficiently high. Otherwise, small/large $\rho_{ag}$ can decrease/increase the optimal cellular contact durations and decrease the ability of the FRAM to recognize rare APC conditions without a significant risk of false positives/negatives. We also note an asymmetry in each panel of Fig 3A–3C and 3G–3I where the accuracy $\Gamma(\tau^*)$ more quickly reduces to the baseline (4) at smaller values of $\rho_{ag}$. This demonstrates that the false positive rate may be more problematic than the false negative rate for T cell classification accuracy.

## High decision accuracy comes at the cost of increased number of energy utilizing reactions

Theoretical works have hypothesized that kinetic proofreading involves energy-consuming reactions [29–33], for example phosphorylations of the TCR complex following TCR/pMHC binding [1, 38, 40]. To estimate such a cost, we approximate the mean number of futile reactions $n_e$, i.e., the forward reactions in the KP mechanism that do not result in TCR activation over a contact duration $\tau$. We estimate this quantity by deriving and solving a similar ODE system to that of which was utilized in the approximation of the extreme statistic (Sec. 5 of S1 Text). We do not count the initial binding event so that the cost to reach the state $C_i$ from state

$C_0$ is $i$ futile reactions. A single TCR/pMHC complex that reaches the $i^{th}$ bound state in the KP mechanism (Fig 1) before dissociating results in $i$ futile reactions ($i < n_{KP}$).

In Fig 4 we observe the relationship between T cell accuracy and energetic reactions by plotting a scaled accuracy $\tilde{\Gamma}$ defined as

$$
\tilde{\Gamma} = \begin{cases} 0, & \Gamma \leq \Gamma^{(BL)} \\ \dfrac{\Gamma - \Gamma^{(BL)}}{1 - \Gamma^{(BL)}}, & \Gamma \geq \Gamma^{(BL)} \end{cases} \qquad \Gamma^{(BL)} = \left| \frac{1}{2} - \rho_{ag} \right| + \frac{1}{2}. \tag{8}
$$

High T cell accuracy ($\tilde{\Gamma} \approx 1$) is associated with a large number of futile reactions, particularly when $\sigma$ and $\rho_{ag}$ are small. As $\sigma$ and/or $\rho_{ag}$ decreases (Fig 4 columns), a larger $n_{KP}$ is needed to achieve moderate or large temporal regions of high accuracy. The increased energy requirement at smaller $\sigma$ and $\rho_{ag}$ values is a result of the increase in the number of KP steps. As $n_{KP}$ increases, a longer contact duration is necessary to capture the first passage activation of an agonist antigen, i.e., observing a true positive activation. This time constraint can be reduced by increasing the KP rates of the FRAM (Sec. 6 of S1 Text). However, if the KP rates are scaled equally, then the number of futile reactions does not change (Sec. 6 of S1 Text).

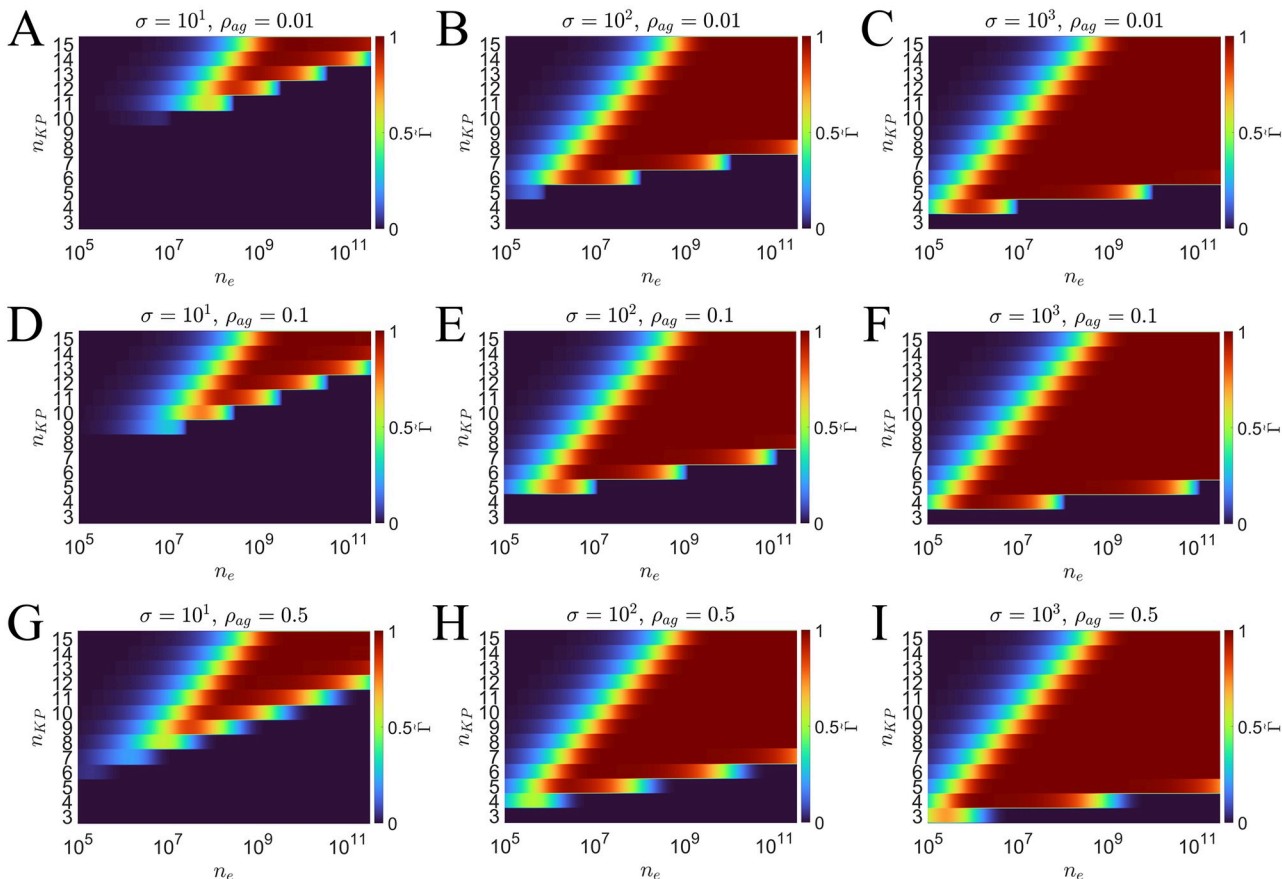

**Fig 4. Plots showing the classification accuracy in terms of the number of energy utilizing reactions ($n_e$) and $n_{KP}$.** The scaled accuracy $\tilde{\Gamma}$ is such that $\tilde{\Gamma} = 0$ is $\Gamma \leq \Gamma^{(BL)}$ and $\tilde{\Gamma} = (\Gamma - \Gamma^{(BL)})/(1 - \Gamma^{(BL)})$. The left, center, and right column of plots show results for $\sigma = 10$, $\sigma = 10^2$, $\sigma = 10^3$, respectively. Each top, center, and bottom row shows results for $\rho_{ag} = 0.01$, $\rho_{ag} = 0.1$, and $\rho_{ag} = 0.5$.

In Fig 4, it is clear that there are regions of the parameter space where increasing the number of KP steps or increasing the number of futile reactions can result in decreased accuracy. This is also true for the T cell/APC contact duration (Sec. 4 of S1 Text), where increasing the cellular contact duration results in a lowered accuracy. These results demonstrate that increasing only one of these model parameters is not necessarily sufficient to increase the accuracy. In these cases, if there is an increase in $n_{KP}$, then there must be an accompanied increase in $n_e$ and/or $\tau$. This follows from the previous results showing that when parameters are held constant, an increase in the number of KP steps requires reevaluation of the optimal cellular contact duration $\tau$, which shifts to larger values. A similar situation occurs if the number of futile reactions, or cellular contact duration, is increased and $n_{KP}$ is held fixed. A logical assumption may be that increasing the number of futile reactions (or energy) in the system would lead to a better APC classifier. However, when $n_{KP}$ is held fixed, the increased number of futile reactions increases the probability of T cell activation for both, agonist and self antigen populations. This means that too much energy in the form of futile reactions (and/or too much contact time) would lead to a high false positive probability.

## Discussion

### Summary of results

We developed a first receptor activation model (FRAM) to evaluate the T cell as an APC classifier with a finite decision time. We used mathematical analysis of extreme statistics to determine the probabilities of T cell activation by either a single agonist antigen or numerous self-antigen. We evaluated the model in a challenging environment such as when self and agonist antigen are similar in KP properties but differ largely in expression or when agonist positive APCs are rare in a population. We used the accuracy (3) to measure T cell activation outcomes over a range of cellular contact durations and environmental conditions in order to investigate the FRAM as a classifier of APC agonist status.

We found that a high classification accuracy can be achieved over a large window of cellular contact times (Fig 2) given a sufficiently large $n_{KP}$ (kinetic proofreading steps) and/or $\sigma$ (ratio of antigen disassociation rates). Outside this window, poor accuracy can arise from contacts that are either too short or too long due to false negatives and positives, respectively. In addition, our results showed that the FRAM could overcome the challenge of similar agonist/self antigen ligands (small $\sigma$) and large disparities in self/agonist expression ($n_{ag} = 1$ and $n_{self} = 10^4$) by sufficiently increasing the number of KP steps (Fig 2C and 2F).

Accurate classification is more challenging when $\rho_{ag}$ is small/large due to a higher false positive/negative rate. Additionally, we found that the agonist positive prevalence can influence the optimal contact duration. When the agonist positive prevalence is small, the false positive rate increases (Fig 3D–3F) which yields shorter optimal contact durations(Fig 3G–3I), since decreasing the contact duration reduces the probability of T cell activation. When the agonist positive prevalence is large, the false negative rate increases which yields longer optimal contact durations, since increasing the cellular contact time increases the probability of activation. Additionally, we found that the accuracy of the FRAM is effectively independent of the agonist positive prevalence when $n_{KP}$ is sufficiently large (Fig 3C, 3F and 3I), even when $\sigma$ is small. This demonstrates that the FRAM can simultaneously overcome the challenge of similar self/agonist ligand as well small agonist positive prevalence by sufficiently increasing the number of KP steps.

Lastly, we quantified the cost in achieving high classification accuracy by considering the number of futile reactions in the FRAM. We found that for smaller $\rho_{ag}$ and/or $\sigma$ values, more futile reactions are necessary to effectively recognize both agonist positive and agonist negative

cells (Fig 4). The primary contributor to the number of futile reactions is the suppression of a large self antigen population in the kinetic proofreading mechanism. When $\sigma$ is large (Fig 4C, 4F and 4I), this suppression is achieved with a smaller number of kinetic proofreading steps, thus reducing the number of futile reactions. When $\rho_{ag}$ is large (Fig 4G–4I), the FRAM is capable of achieving good classification accuracy with smaller numbers of $n_{KP}$ since the false positive risk is low. We found that the most difficult case is when both, $\rho_{ag}$ and $\sigma$ are small Fig 4A. In this case, a large $n_{KP}$ is necessary to successfully identify the rare agonist positive cells while remaining insensitive to the large proportion of agonist negative cells. This has the added effect of requiring a large number of futile reactions, since most encounters are with an agonist negative APC and the FRAM must be able to suppress receptor triggering from the numerous self antigen in each of these encounters.

## Conclusions

We found that the FRAM is capable of high classification accuracy when agonist and self antigen are similar and differ largely in expression within a T cell/APC contact (Fig 2C and 2F). In addition, we found that our model could achieve this high accuracy even when agonist positive APCs are rare (Fig 3C, 3F and 3I). This suggests that in terms of first passage times of agonist and self antigen, kinetic proofreading is more capable of antigen discrimination than what has been observed in T cell experiments [9]. Furthermore, unlike previous works [28, 30, 61], the FRAM yields no trade-off between remaining sensitive to agonists and being able to distinguish between agonist and self antigen populations.

Our results indicate that while the FRAM is capable of high levels of accuracy, it may be associated with certain costs, or biological constraints. We showed that when agonist and self antigen are more alike (smaller $\sigma$), a larger $n_{KP}$ is needed (Fig 2C), which increases the cellular contact duration necessary to capture the first passage activation of an agonist antigen (Sec. 4 of S1 Text). Additionally, we found that when agonist positive cells are rare, more $n_{KP}$ are necessary to successfully suppress the false positive rate (Fig 3D–3F), which again has the effect of increasing the optimal cellular contact duration (Fig 3I). This may suggest that the cellular contact duration could act as a cost for accurate T cell responses.

From a mathematical perspective, we can scale the cellular contact durations to any value by increasing the rates of the model (Sec. 6 of S1 Text). However, this can yield large reaction rates ($k_p > 10^9 s^{-1}$ in Fig 3I) which may also have a biological constraint. We found that this method of scaling has no influence on the number of futile reactions (Sec. 6 of S1 Text), which are reactions that may require energy consumption. Just as with the cellular contact duration, we found that more futile reactions are necessary for accurate APC recognition as $n_{KP}$ increases (Fig 4). Hence, our results also demonstrate how energetic costs may act as a constraint in T cell antigen discrimination. These potential costs may provide insight for observations in biological experiments in which large discrepancies are observed in dissociation rates between agonist and self antigen [41]. Additionally, this may further support the idea that kinetic proofreading alone is not sufficient to explain some of the observations in previous T cell antigen discrimination experiments [9–15, 36].

In conclusion, we have shown how viewing the classic kinetic proofreading mechanism through resilience to extreme statistics (self activation) offers a different perspective on the problem of antigen discrimination. By modeling T cell activation as a first passage time describing receptor triggering by agonist or self antigen in the kinetic proofreading mechanism, we were able to show that the FRAM was capable of near perfect accuracy in challenging environmental conditions. We also showed several potential costs that may act as biological constraints to high accuracy in the T cell environment. Our hope is that the simplicity of this

model yields a foundation from which more complexity can be built with respect to observed T cell characteristics, such as $Ca^{2+}$ signaling and microvilli structures.

## Supporting information

**S1 Text. Mathematical details and convergence analysis.**
(PDF)

## Author Contributions

**Conceptualization:** Jonathan Morgan, Alan E. Lindsay.

**Data curation:** Alan E. Lindsay.

**Formal analysis:** Jonathan Morgan, Alan E. Lindsay.

**Funding acquisition:** Alan E. Lindsay.

**Investigation:** Alan E. Lindsay.

**Methodology:** Jonathan Morgan, Alan E. Lindsay.

**Software:** Jonathan Morgan.

**Supervision:** Alan E. Lindsay.

**Validation:** Jonathan Morgan.

**Writing – original draft:** Jonathan Morgan, Alan E. Lindsay.

**Writing – review & editing:** Jonathan Morgan, Alan E. Lindsay.

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
