## [Decision Letter · Decision Letter 0]

17 Jul 2023

Dear Lindsay,

Thank you very much for submitting your manuscript "Modulation of antigen discrimination by duration of immune contacts in a kinetic proofreading model of T cell activation with extreme statistics." for consideration at PLOS Computational Biology. As with all papers reviewed by the journal, your manuscript was reviewed by members of the editorial board and by several independent reviewers. The reviewers appreciated the attention to an important topic. Based on the reviews, we are likely to accept this manuscript for publication, providing that you modify the manuscript according to the review recommendations.

The reviewers appreciate the novelty and insights gained from your work and suggest edits to clarify several points, largely regarding modeling details.

Sincerely,

Inna Lavrik

Academic Editor

PLOS Computational Biology

Stacey Finley

Section Editor

PLOS Computational Biology

Reviewer's Responses to Questions

**Comments to the Authors:**

Reviewer #1: T cells form contacts with antigen presenting cells in order to make a binary decision between whether or not to mount an immune response. This decision is made with single-molecule sensitivity and high specificity. A well-established model explaining this specificity and sensitivity is kinetic proofreading, but the present authors point out that it is alone insufficient to explain the T cell's performance. Here, the authors add a finite lifetime of a cell-cell contact to the kinetic proofreading model, and find that this enhances the accuracy. They achieve this through use of extreme statistics, analyzing rare events rather than solely relying on mean-field approximations.

The computational and mathematical results in this paper contribute a novel, unappreciated role for the lifetime of the cell-cell contact. The authors clearly and rigorously give background for the need of their new model, both biologically and mathematically. The idea of terminating cell-cell contacts leading to one hundred percent accuracy is an unexpected and powerful result.

The following comments are minor.

SPECIFIC COMMENTS

1. For some model parameters, estimates from previous literature are discussed, but not the number of KP steps. Since this is such a key parameter, the authors should include discussion of other estimates of the number of KP steps. From a molecular structural viewpoint: how many phosphosites are there per TCR? How many ITAMs? This gives a rough upper bound on the number of steps. From a whole-cell viewpoint: Pettman et al. 2021 eLife also contains related results.

2. Figure 4 and the subsection on energy is unclear. Indeed, it seems the results shown in the figure contradict the headline: There are regions of parameter space where reducing contact lifetime allow the consumption of less energy, while increasing the accuracy.

3. Is the fact that T_1,nself is a Weibull random variable used in the simulations? In other words, do the simulations sample from a Weibull to accelerate simulation time? If so, add details in the model section.

4. Typo: First paragraph of results: sigma ≥ 10 should be replaced with n_KP ≥ 10

5. Figure 2B appears to have strange bump in the sigma=15 curve near the peak. Is this a rendering error, or a numerical artefact?

Reviewer #2: In this paper, the authors present and analyze a stochastic mathematical model of antigen discrimination that builds on a kinetic proofreading paradigm. Compared to prior theoretical work, the model does not make a steady-state approximation and in particular allows the authors to investigate how contact time duration (which is known to vary widely) affects the accuracy of the antigen discrimination mechanism. The authors also investigate how varying other parameters (such as number of kinetic proofreading steps and how the dissociation rates differ between self and agonist antigen) affect the accuracy of this mechanism. The authors find that this mechanism can be highly accurate (in terms of a small false positive probability and a small false negative probability) in a variety of parameter regimes if the contact duration is appropriately chosen. Mathematically, the model consists of continuous-time Markov chains and the authors must estimate a certain first passage time.

I think this is a very interesting paper that deserves to be published in PLOS Comp Bio. The authors address an important biological problem which has been studied with theoretical models for quite some time, and yet the authors present an important new perspective that prior ``steady-state'' analyses seem to miss. My main critiques are the following more technical questions which I think the authors should address. I think most of these critiques could be addressed by a clearer description of the model in the main text.

-Page 6 of the main text says that the formula for the accuracy $\\Gamma$ is given in the supplement. Where is the formula for $\\Gamma$ in the supplement? I couldn't find it.

-In some instances, the authors find optimal values of $\\tau$ are on the order of $\\tau=10^{5}$. Looking at Table 1, $\\tau$ is said to have units of seconds, in which case this is $\\tau\\approx30$ hours which I assume is unphysiological. By saying that $\\tau$ has units of seconds, do the authors merely mean that it has units of time, and the authors have simply scaled time for that $k_{1}$, $k_{p}$, and $k_{-1}$ are all unity (along the lines of section 5 in the supplement)? If so, can the authors estimate the values of these rates (from the literature) to translate their optimal values of $\\tau$ into real time values? Along these lines, are there available estimates for $\\sigma$?

-Are the units of $k_{1}$ correct? Table 1 says that it is an inverse time, but looking at equation (14a) in the supplement, it seems like it is a bimolecular reaction rate and thus has units of $1/(concentration*time)$ assuming $L_{T}$ has units of concentration. It says $L_{T}$ is the total population of ligands, so the units are simply the raw number of ligands? Perhaps $R_{T}$, $L_{T}$ should be added to Table 1?

-The supplement presents a few different approximations. Which one of these approximations is used to plot $\\Gamma$ in the main text figures?

-I think the ODE approximation in the supplement should be clarified. In particular, I don't understand the ``nonlinear binding rate $K_{1}(R(t),L(t))$'' mentioned at the top of page 5 of the supplement.

-The self-activation time is defined as the minimum of $t_{1}, t_{2}, \\dots t_{n_{self}}$. Are these random variables independent and identically distributed (iid)? I think they are, but then I don't understand the following statement from the Supplement: ``A source of numerical error in applying this result [3] to a large antigen population in the FRAM is that the first passage activation of receptors are not i.i.d., since the propensity for binding events is Kon = konRL.'' I don't understand this sentence. I think this sentence should be clarified and the model assumptions (namely of independence or dependence) should be clarified in the main text.

-On page 4 of the supplement, there are some references to equation (13) which should be references to equation (12) (for example, it says ``The integral (13) is readily...''

**Have the authors made all data and (if applicable) computational code underlying the findings in their manuscript fully available?**

Reviewer #1: Yes

Reviewer #2: Yes

PLOS authors have the option to publish the peer review history of their article (what does this mean?). If published, this will include your full peer review and any attached files.

Reviewer #1: No

Reviewer #2: No

Figure Files:

Data Requirements:

Reproducibility:

References:

---

## [Editor Report · Decision Letter 1]

5 Aug 2023

Dear Lindsay,

We are pleased to inform you that your manuscript 'Modulation of antigen discrimination by duration of immune contacts in a kinetic proofreading model of T cell activation with extreme statistics.' has been provisionally accepted for publication in PLOS Computational Biology.

Best regards,

Inna Lavrik

Academic Editor

PLOS Computational Biology

Stacey Finley

Section Editor

PLOS Computational Biology

---

## [Editor Report · Acceptance letter]

23 Aug 2023

PCOMPBIOL-D-23-00838R1 

Modulation of antigen discrimination by duration of immune contacts in a kinetic proofreading model of T cell activation with extreme statistics.

Dear Dr Lindsay,

I am pleased to inform you that your manuscript has been formally accepted for publication in PLOS Computational Biology. Your manuscript is now with our production department and you will be notified of the publication date in due course.

With kind regards,

Zsofi Zombor
